# Effects of Surface Roughness on the Electrochemical Properties and Galvanic Corrosion Behavior of CFRP and SPCC Alloy

**DOI:** 10.3390/ma13184211

**Published:** 2020-09-22

**Authors:** YoungHwan Kim, MyeongHan Yoo, MinSeok Moon

**Affiliations:** 1Technical Engineering Team, High Engine Ltd., 18, Chopodari-ro, Deokjin-gu, Jeonju 54817, Korea; ddd737373@hanmail.net; 2Composite Engineering Center, Korea Institute of Carbon Convergence Technology, #110-11 Banryong-ro, Jeonju 54853, Korea; anrkr@kctech.re.kr

**Keywords:** carbon fiber reinforced plastics (CFRP), SPCC alloy, surface treatment, salt test condition, variation in microstructure, variation in chemical composition, potentiodynamic corrosion test, galvanic corrosion test

## Abstract

This study investigated the potentiodynamic corrosion behavior of carbon fiber reinforced plastic (CFRP) and automotive rolled mild steel alloy (SPCC alloy) under different surface roughness conditions. Electrochemical characterization was performed using a potentiodynamic corrosion test with 5.0 wt.% NaCl aqueous solution at 25 ± 2 °C, while microstructural and compositional changes before and after corrosion were evaluated using field emission scanning electron microscopy (FE-SEM) and energy dispersive spectroscopy (EDS), respectively. The CFRP and SPCC corrosion rate increased as surface roughness increased. Generally, SPCC corroded faster than CFRP. The surface composition of CFRP was not affected by corrosion, regardless of the surface roughness conditions. Conversely, SPCC exhibited remarkable changes due to the formation of oxides, and its corrosion was more severe than that of CFRP as surface roughness increased. We used a double flat electrode cell to conduct a galvanic corrosion test in this study at 25 ± 2 °C. In this galvanic corrosion test, we studied different kinds of surface roughness for SPCC specimens under the CFRP material in its as-received condition and #200 condition. We confirmed that the results of galvanic corrosion for this study have a difference in corrosion amount and corrosion rate of SPCC specimens according to the surface roughness of CFRP.

## 1. Introduction

The rapid development of technologies for exhaust pollution reduction in the automobile industry has been accompanied by a growing consumer demand for increased engine output and reduced fuel consumption. In addition, eco-friendly materials, aluminum (Al) alloys, and magnesium (Mg) alloys have been incorporated into new automotive parts in various ways to minimize human health risks. Lightweight materials like Al, Mg, and carbon fiber reinforced plastic (CFRP) can help improve the fuel consumption of vehicles. These materials have already been commercialized as major alternative parts in high-end vehicles. The preference for lightweight vehicle parts has led to a demand for new materials that meet various consumer demands, including safety and convenience [1,2,3]. The evaluation of vehicle performance improvement and long-term durability of vehicle parts relies on precise observations of the material’s corrosion behavior. Lightweight materials containing Al, Mg, and CFRP have recently garnered considerable attention. However, the practical implementation of these materials has been limited by the degradation of their strength, durability, and reliability due to corrosion associated with dissimilar material in joining technologies [4,5,6].

This study investigated lightweight CFRP materials and an automotive rolled mild steel alloy (known as SPCC). The electrochemical properties of the samples with varying surface conditions were evaluated via a salt test (5.0 wt.% NaCl aqueous solution). The microstructural and compositional changes after electrochemical evaluation were analyzed to establish the applicability of CFRP and SPCC as multi-materials in automotive parts [7,8,9,10].

## 2. Materials and Methods

### 2.1. Materials

Commercial 4k and 10-layer polished plain weave CFRP and automotive SPCC were purchased from an American multinational e-commerce corporation, eBay website (San Jose, CA, USA), and used. The chemical compositions of CFRP and SPCC are presented in Table 1 [11].

### 2.2. Fabrication of Specimens for Each Surface Condition

CFRP and SPCC specimens with random surface conditions were obtained using #200 and #2000 sandpapers (Deer, Deerfos Co., Ltd., Seoul, Korea); the roughest surface was achieved using the #200 sandpaper. Consequently, there were three groups of specimens with varying surface roughness: as-received specimens, #200 sandpaper-abraded specimens, and #2000 sandpaper surface-polished specimens. Each surface condition group consisted of three specimens. Surface polishing was performed for 30 s at a rate of 180 rpm (Metpol, R&B Co., Ltd., Daejeon, Korea) in the presence of water. Both sides of each specimen were polished under the same conditions. The polished specimen was cleaned in anhydrous ethanol using an ultrasonic cleaner for 3 min.

### 2.3. Surface Roughness

The surface roughness values of the as-received and surface-polished specimens were measured (SJ-400, Mitutoyo Co., Ltd, Kawasaki, Japan); the Ra value was used as the measured value. The Ra value is the mean center line roughness. The center of each specimen was measured thrice, and the values obtained from these measurements were averaged.

### 2.4. Potentiodynamic Corrosion Properties

The electrochemical properties, i.e., the direct corrosion behavior, of the as-received and surface-polished specimens were evaluated via potentiodynamic anodic polarization testing (PARSTAT MC, AMETEK, Oak Ridge, TN, USA). A 5.0 wt.% NaCl aqueous solution served as the electrolyte, and represented the salt test conditions for automobiles [12]. An Ag/AgCl electrode (Ag·AgCl/KCl, 3.5 M) was used as the reference electrode. The measurement voltage ranged from −0.25 to 1.20 V (Ag·AgCl) at a scan rate of 1.0 mV/sec at 25 ± 2 °C. The generated gas was naturally discharged because a separate gas discharge device was not used.

### 2.5. Microstructure and Surface Composition

The microstructure of each sample’s comparison of before and after potentiodynamic anodic polarization testing was analyzed 50 times with 1000 times magnification using the field emission scanning electron microscopy (FE-SEM; JSM- 7100F, JEOL Ltd., Tokyo, Japan).

Furthermore, the chemical composition of each sample’s comparison of before and after potentiodynamic anodic polarization testing was simultaneously evaluated using energy dispersive spectroscopy (EDS; X-Max, Oxford Instruments, Abingdon, UK).

### 2.6. Galvanic Corrosion

Galvanic corrosion evaluation was performed using electrochemical measurement equipment (PARSTAT MC, AMETEK, Oak Ridge, TN, USA). In this study, a galvanic corrosion test was performed on the surface conditions of all SPCC materials based on the CFRP as-received test piece and #200 test piece using the double flat electrode cell. A double flat electrode cell (F031, AMETEK, Oak Ridge, TN, USA) was used with a 100 mL 5.0 wt.% NaCl aqueous solution as the electrolyte. This double flat electrode cell experiment provides different information compared to the common galvanic corrosion test experiment [7,9]. This experiment was conducted on a galvanic corrosion test instrument without directly connecting the CFRP and SPCC specimens. The working electrode area was 10 mm^2^. The galvanic corrosion test was conducted for 200 s at 25 ± 2 °C.

## 3. Results and Discussion

### 3.1. Surface Condition

The surface roughness of the specimens was compared to the values of the Korea Standard (KS) B 0161 specification [13]. The surface roughness of the as-received CFRP corresponded to the value of N2 (0.04 µm), the #2000 surface-polished to N3 (0.11 µm), and the #200 surface-polished to N6 (0.80 µm) (Figure 1) [13,14]. The as-received SPCC corresponded to the value of N5 (0.34 µm), the #2000 surface-polished to N5 (0.43 µm), and the #200 surface-polished to N7 (1.36 µm) [13,14]. The as-received SPCC specimen was formed via cold rolling; therefore, its surface roughness value was dependent on the surface condition of the roller used during cold rolling. The random surface conditions of the #200 and #2000 surface-polished SPCC specimens were determined using sandpaper, and the Ra value, in turn, depended on the type of sandpaper used.

### 3.2. Potentiodynamic Corrosion Properties

The electrochemical properties of the CFRP and SPCC specimens, with varying surface roughness, after conducting the potentiodynamic corrosion tests, are summarized in Table 2.

The as-received CFRP specimen exhibited a corrosion current density (I_CORR._) of 4.547 × 10^−11^ A/cm^2^ during potentiodynamic corrosion testing, that increased to 5.397 × 10^−8^ and 7.691 × 10^−7^ A/cm^2^ in the #2000 and #200 surface-polished specimens, respectively (Figure 2). Because the corrosion current density is related to the corrosion speed, the CFRP specimens corroded faster as the surface roughness increased. Furthermore, the coarse particles of the #200 sandpaper exposed some of the carbon fibers during polishing; these exposed fibers reacted with the electrolyte and underwent rapid oxidation-reduction.

Therefore, in this study, the corrosion rate of the CFRP specimens increased, with an increase in surface roughness, thereby expanding the reaction surface area.

The corrosion current density of the as-received SPCC specimen was 7.986 × 10^−3^ A/cm^2^, which increased slightly to 8.866 × 10^−3^ and 1.039 × 10^−2^ A/cm^2^ for the #2000 and #200 surface-polished specimens, respectively (Figure 3). Although the surface roughness tendencies of the SPCC specimens were similar to those of the CFRP specimens, surface roughness had a negligible effect on the corrosion current density.

Therefore, all types of SPCC specimens exhibited significantly higher corrosion speeds than CFRP. The SPCC specimen, being a metallic material, was easily dissolved and oxidized in the 5.0 wt.% NaCl aqueous solution compared with CFRP, a non-metallic material [15].

### 3.3. Microstructure and Component Analysis

The microstructural analysis revealed exposed carbon fibers on the #200-sandpapered CFRP specimen with the formation of haphazard surface roughness (Figure 4, yellow circle indicated). All the CFRP specimens only exhibited slight microstructural changes after corrosion testing. Furthermore, negligible changes in the chemical composition of the specimens were observed (Table 3).

These results indicated that regular CFRP materials are covered with polymer materials, particularly carbon fiber, as a passivation layer. However, the exposed carbon fiber, due to sandpapering using #200 emery paper in this experiment, was related to an increased corrosion rate (Figure 2). In this experiment, the exposed carbon fiber acted as a conductor to electrons in the electrolyte solution, thereby increasing corrosion speed.

The SPCC specimens exhibited significant microstructural changes after corrosion (Figure 4). Consequently, various oxides were generated on the surface during corrosion (Table 2).

The chemical compositions of CFRP and SPCC are given in Table 1. Nevertheless, the analysis of the chemical composition of SPCC after corrosion confirmed the existence of other metallic trace elements.

In this experiment, the fine metal elements of the SPCC specimen detected after the potentiodynamic corrosion test were expected to be elements that existed in the iron ore raw material in the melting of iron ore. However, because no trace elements were detected through the spectroscopic analysis of pristine SPCC, the elements have not been specified.

Metallic trace elements were not detected in commercial SPCC via EDS; this can be explained by the presence of a stable Fe_2_O_3_ film on the SPCC surface.

After the potentiodynamic corrosion experiment, metallic trace elements were detected in SPCC through EDS analysis.

Overall, CFRP exhibited almost no corrosion tendency in 5.0 wt.% NaCl aqueous solution, regardless of its surface condition. However, the SPCC specimens corroded more severely owing to oxidation, as the surface roughness increased.

### 3.4. Galvanic Corrosion

Galvanic corrosion in different metallic materials is generally measured using a coupling contact by bolting or welding under the same electrolytic conditions. Furthermore, galvanic corrosion has been primarily investigated for metallic materials. However, in recent years, as the use of composite materials such as CFRP, a non-metallic material, has increased, the investigation of galvanic corrosion between metallic-nonmetallic coupled materials has garnered interest [7,9].

In this study, unlike in previous studies, the non-metallic CFRP and metallic SPCC did not have coupled contact conditions. Furthermore, in this study, a galvanic corrosion evaluation was performed individually for each of the specimens that exhibited different surface roughness, in the same electrolyte using a double flat electrodes cell.

The galvanic corrosion of the as-received CFRP specimen (with the most intact surface) and the #200-sandpapered CFRP specimen (with the roughest surface) was compared with the galvanic corrosion of the corresponding SPCC specimens (Figure 5 and Figure 6). The as-received SPCC specimen corroded less than the other SPCC specimens, where the #200-sandpapered SPCC specimen exhibited the largest corrosion on the as-received CFRP condition. All the SPCC specimens exhibited a gradual increase in corrosion rate as time elapsed.

Furthermore, the as-received SPCC specimen exhibited the least corrosion on the #200-sandpapered CFRP condition, while the #200-sandpapered SPCC specimen corroded the most. The corrosion rate of all the SPCC specimens increased continuously with time.

This study examined the galvanic corrosion tendency of CFRP and SPCC specimens with different surface roughness. Many papers have reported galvanic corrosion in electrolytes, typically after coupling the two materials. CFRP and SPCC specimens with three types of surface roughness were compared; as-received CFRP had the slowest corrosion rate (Figure 5). As a result, the corrosion of the as-received specimen was established as the lowest among the SPCC specimens, as indicated in Figure 5 and Figure 6; the corrosion reaction increased as the corrosion time increased to 200 s. Corrosion of the #2000 SPCC specimen was found to increase as corrosion progressed. The #200-sandpapered SPCC specimen underwent the most corrosion, and the corrosion rate increased slightly to 175 s.

Three types of SPCC and CFRP surface condition specimens were compared. The galvanic corrosion of the #200 CFRP was found to have the fastest corrosion rate among all the CFRP specimens (Figure 6). The results also indicated that the extent of corrosion of the as-received SPCC specimen was the lowest among all the SPCC specimens and corrosion increased as the galvanic corrosion reaction progressed. The #2000- and #200-sandpapered SPCC specimens showed similar galvanic corrosion behavior, and the #200-sandpapered SPCC specimen was found to have undergone the most extensive corrosion.

These results established that the corrosion rate and amount of corrosion were changed through galvanic corrosion based on the surface conditions of the CFRP and SPCC specimens, and that the rougher the surface of the CFRP specimens, the higher the corrosion current density.

## 4. Conclusions

The electrochemical and microstructural properties of lightweight CFRP and SPCC automotive materials were compared before and after corrosion via salt test conditions (5.0 wt.% NaCl aqueous solution); the effect of surface roughness was particularly considered. The following conclusions were made:(1)The corrosion current density of CFRP increased with increasing surface roughness, while SPCC only exhibited a slight increase in corrosion rate as surface roughness increased. Therefore, SPCC metallic materials with varying surface roughness were easily corroded, based on the potentiodynamic values obtained in the presence of a 5.0 wt.% NaCl aqueous solution. Furthermore, CFRP non-metallic materials exhibited significantly lower corrosion speeds compared with SPCC metallic materials.(2)CFRP specimens exhibited no change in microstructure and surface composition after corrosion in a 5.0 wt.% NaCl aqueous solution, regardless of the surface roughness. However, SPCC specimens exhibited a dramatic change in microstructural and compositional properties with an increase in surface roughness. An analysis of the surface composition of the SPCC specimens revealed a distinct increase in oxygen content after the potentiodynamic corrosion test in 5.0 wt.% NaCl aqueous solution, due to the oxidation of some elements in the SPCC alloy (the elements Cr, Ti, P, and S were presumed to be introduced during the iron ore melting process). In this experiment, the SPCC specimens, through electrochemical reactions, generated many oxide species. Hence, it was confirmed that steel materials, being metallic, have a higher corrosion rate than CFRP, a non-metallic material, under salt test conditions.(3)In the galvanic corrosion results for a double flat electrodes cell test, the as-received SPCC specimen exhibited less galvanic corrosion than the surface-polished SPCC specimens, regardless of the condition of CFRP. The as-received SPCC specimen has a stable oxide film (Fe_2_O_3_) on the surface. Moreover, the galvanic corrosion results revealed that the corrosion rate of all the SPCC specimens gradually increased with increase in the corrosion time, compared with the CFRP specimens. Therefore, as-received steel materials are expected to exhibit a slow corrosion rate that increases over time in 5.0 wt.% NaCl aqueous solutions. These results offer insight into the improvement of long-term strength, durability, and reliability of dissimilar material joints of multi-material automotive systems, which involve CFRP and SPCC.

## Figures and Tables

**Figure 1 materials-13-04211-f001:**
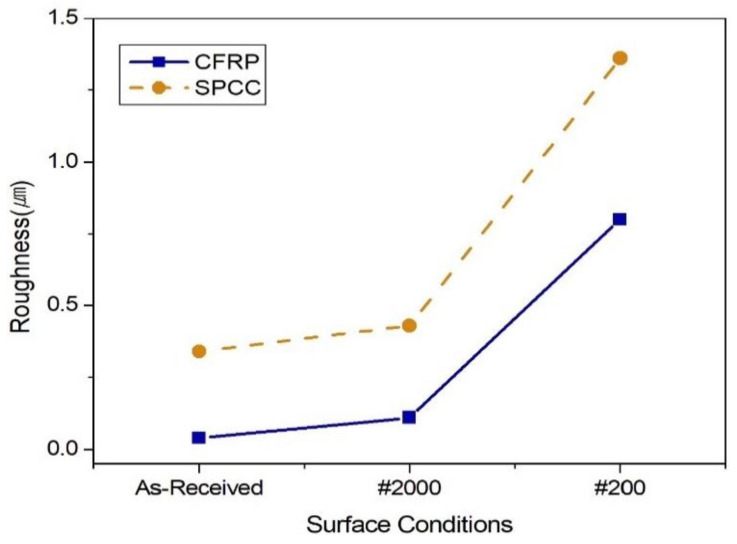
Surface roughness of CFRP (orange) and SPCC (blue).

**Figure 2 materials-13-04211-f002:**
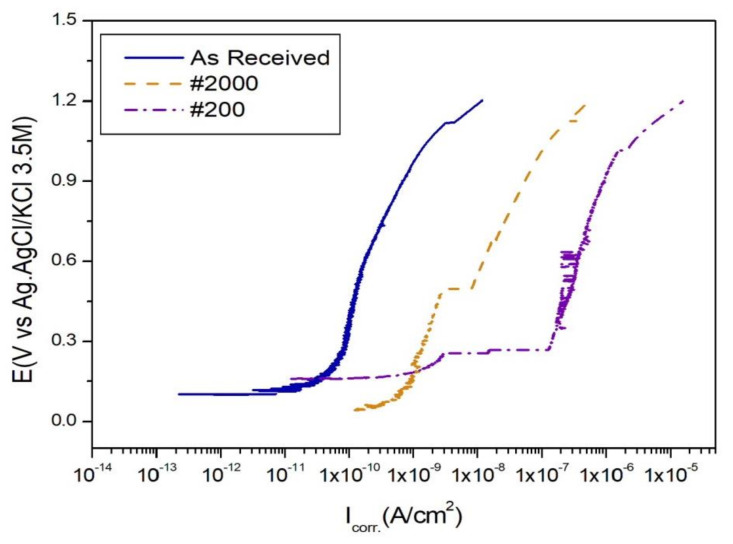
Corrosion current density of CFRP during potentiodynamic corrosion testing with a 5.0 wt.% NaCl aqueous solution electrolyte.

**Figure 3 materials-13-04211-f003:**
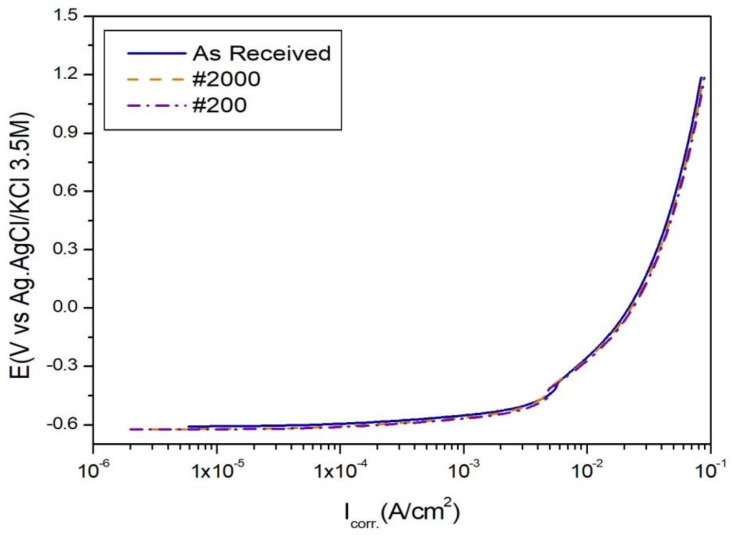
Corrosion current density of SPCC during potentiodynamic corrosion testing with a 5.0 wt.% NaCl aqueous solution electrolyte.

**Figure 4 materials-13-04211-f004:**
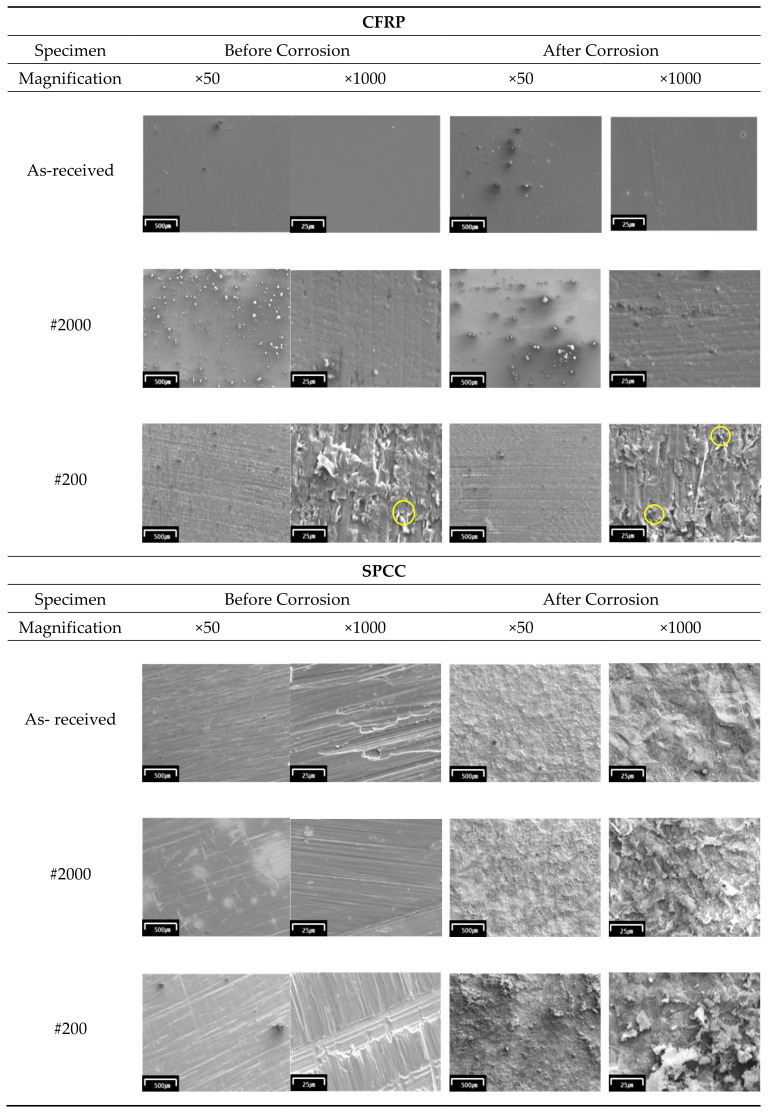
Microstructure of CFRP and SPCC, before and after corrosion testing.

**Figure 5 materials-13-04211-f005:**
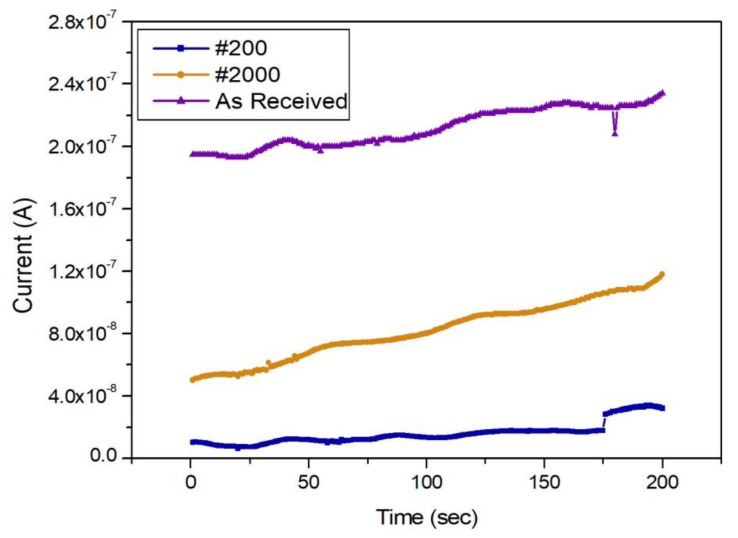
Graph of galvanic corrosion results at all SPCC surface conditions for as-received CFRP.

**Figure 6 materials-13-04211-f006:**
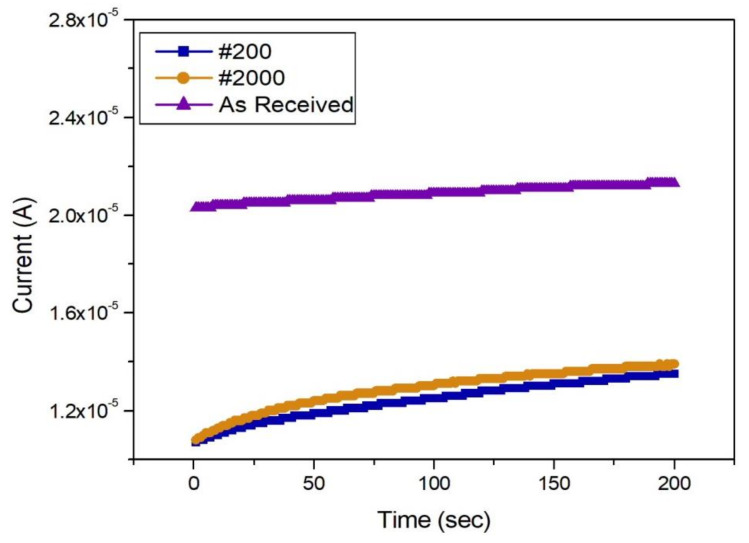
Graph of galvanic corrosion results at all SPCC surface conditions for #200 CFRP.

**Table 1 materials-13-04211-t001:** Chemical compositions of carbon fiber reinforced plastic (CFRP) and SPCC alloy obtained from EDS analyses.

Materials	Elements (wt.%)						Remarks
CFRP	O	C					EDS analysis
	17.24	82.76					
SPCC	O	C	Mn	P	S	Fe	EDS analysis
	1.40	-	-	-	-	98.60	
	O	C	Mn	P	S	Fe	Spec.
	-	<0.15	<0.60	<0.050	<0.050	Bal.	

**Table 2 materials-13-04211-t002:** Summary of potentiodynamic corrosion results for the CFRP and SPCC specimens with varying surface roughness.

Specimen	Surface Conditions	E_corr._(V)	I_corr._(A/cm^2^)	β_A_ * (V/dec.)	β_C_ *(V/dec.)
CFRP	As Received	4.234 × 10^−1^	4.547 × 10^−11^	7.779 × 10^−2^	9.316 × 10^−2^
	#2000	3.243 × 10^−1^	5.397 × 10^−8^	1.049 × 10^−1^	7.607 × 10^−1^
	#200	3.767 × 10^−1^	7.691 × 10^−7^	5.739 × 10^−1^	1.298
SPCC	As Received	3.557 × 10^2^	7.986 × 10^−3^	1.321	2.054
	#2000	7.251 × 10^2^	8.866 × 10^−3^	1.397	2.200
	#200	1.557 × 10^−1^	1.039 × 10^−2^	1.637	2.627

* β_A_ = Anodic Tafel constants, β_C_ = Cathodic Tafel constants.

**Table 3 materials-13-04211-t003:** Surface composition of CFRP and SPCC, before and after corrosion testing.

CFRP				
Specimen	Before Corrosion		After Corrosion	
As-received	Element	wt.%	Element	wt.%
	C	82.48	C	81.5
	O	17.52	O	18.5
	Total:	100.00	Total:	100.00
#2000	Element	wt.%	Element	wt.%
	C	85.68	C	84.75
	O	14.32	O	15.25
	Total:	100.00	Total:	100.00
#200	Element	wt.%	Element	wt.%
	C	87.29	C	88.78
	O	12.73	O	11.22
	Total:	100.00	Total:	100.00
**SPCC**				
Specimen	Before Corrosion		After Corrosion	
As-received	Element	wt.%	Element	wt.%
	O	1.4	O	5.3
	Fe	98.6	S	0.84
	Total:	100.00	Ti	2.14
	-		Cr	0.26
	-		Fe	81.97
	-		Cu	5.47
	-		As	0.72
	-		Zr	1.91
	-		Nb	1.39
	-		Total:	100.00
#2000	Element	wt.%	Element	wt.%
	O	2.72	O	20.55
	Cr	0.09	P	1.19
	Fe	97.29	S	1.54
	Total:	100.00	Ti	5.02
	-		Cr	0.55
	-		Fe	59.37
	-		Cu	7.13
	-		As	1.5
	-		Nb	3.15
	-		Total:	100.00
#200	Element	wt.%	Element	wt.%
	O	1.89	O	24.12
	Fe	98.11	P	0.73
	Total:	100.00	S	1.68
	-		Ti	4.36
	-		Cr	0.29
	-		Fe	58.81
	-		Cu	6.79
	-		Nb	3.21
	-		Total:	100.00

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
