# Peer review of "Effects of Surface Roughness on the Electrochemical Properties and Galvanic Corrosion Behavior of CFRP and SPCC Alloy"

_materials, 2020, doi:10.3390/ma13184211_

Round 1
Reviewer 1 Report
Manuscript [Materials]
Review on Manuscript ID: materials-916105
Title: "Effects of surface roughness on the electrochemical properties and galvanic corrosion behavior of CFRP and SPCC in automotive steel materials", authors: YoungHwan Kim, MyeongHan Yoo and MinSeok Moon.
The authors studied the galvanic corrosion of CFRP and SPCC couple in NaCl solution.
The main revisions that I consider to be made are listed below:
*The article requires a rigorous and complete English review.
*This work is partially supported by experimental results and scientific research methods and the authors do not discuss the obtained results and do not compare their data with literature.
*Section 2.1. Materials must contain additional data on the chemical composition of CFRP.
*The authors must highlight the novelty of this study by comparing their own results with results obtained in similar cases. Please add more references.
*The authors must improve the discussions on Figures 2, 3, 4, 5 and 6.
*Figures 2, 3. The explanations are so confuse! The authors used the Tafel equations and consequently calculate the corrosion current densities? Which are current densities and which are corrosion current densities?
*Figures 2, 3 can be represented as a single figure 2 (a and b).
*The experimental data presented in table 2 must be interpreted and explained accordingly. CFRP exhibits poor electrical conductivity but its surface presents increasing and/or decreasing oxygen content before/after corrosion! WHY? SPCC composition after corrosion indicates the presence of Ti, Cr, Cu, As, Zr, Nb? How is it possible that these elements appear as a result of corrosion processes? If the steel contains these elements then table 1 do not corresponds to the reality.
*Figures 5, 6 can be represented as a single figure 4 (a and b).
*3.4. Section refers to galvanic corrosion. Figures 5 and 6 represents the chronopotentiograms of CFRP-SPCC couple and consequently indicate the Icouple variation with time (200 s). How did the authors determine the ”corrosion amount” or ”corrosion rate” from these graphs?
*Some of the references are not written in accordance with the journal's requirements.
*There are many grammar and typing mistakes. The authors must revise the entire manuscript.
Author Response
We have studied the potentiodynamic corrosion for CFRP and SPCC's different surface roughness conditions, and we have galvanic corrosion of CFRP and SPCC with a flat cell system in NaCl solution. And then finally, we have a review of the microstructure and chemical composition variation between potentiodynamic corrosion behaviors.
We thank all the reviewers for the constructive feedback on our manuscript. We have integrated all the suggestions and revised the manuscript accordingly. We hope that the refined manuscript will be considered for publication.
We have sorry for the reviewers. As we revised the paper, there was a need for a title change, so in “Effects of surface roughness on the electrochemical properties and galvanic corrosion behavior of CFRP and SPCC in automotive steel materials,” We would like to change to "CFRP and SPCC alloy." We ask for reviewers' understanding of this.
We would attach to the revised paper for your recommendations.
* The article requires a rigorous and complete English review.
We have considered English review services by Editage (https://www.editage.co.kr/).
*This work is partially supported by experimental results and scientific research methods and the authors do not discuss the obtained results and do not compare their data with literature.
We have revised this paper's results and conclusions based on the reviewer’s comment.
*Section 2.1. Materials must contain additional data on the chemical composition of CFRP.
Based on the suggestion, the results of the EDS component analysis of CFRP used in this study were entered and corrected in Table 1.
*The authors must highlight the novelty of this study by comparing their own results with results obtained in similar cases. Please add more references.
Based on the suggestion, references 7 and 9, corresponding to galvanic corrosion, have been added to the manuscript. The corrosion test results and the results of electrolytic concentration were evaluated and compared with the results in these two references.
In this paper, I have analyzed the potentiodynamic corrosion of CFRP and SPCC materials under different surface roughness conditions by a double flat electrodes cell. In the case of galvanic corrosion of the as-received CFRP specimen and #200 specimen was performed by exposing SPCC to different surface roughness conditions in the same electrolyte through a double flat electrodes cell.
The tendency of corrosion change between the two materials was analyzed based on the surface roughness conditions.
This paper's galvanic corrosion test adopts that unlike previous galvanic corrosion studies, the use of a double flat electrodes cell analyzes the galvanic corrosion characteristics between two materials without the direct coupling of the specimens. This experiment has as the galvanic corrosion characteristics between two materials can be confirmed in a short time, the experiment time for galvanic corrosion evaluation can be reduced.
*The authors must improve the discussions on Figures 2, 3, 4, 5 and 6.
The explanations of Figures 2, 3, 4, 5, and 6 have been provided in the manuscript.
*Figures 2, 3. The explanations are so confuse! The authors used the Tafel equations and consequently calculate the corrosion current densities? Which are current densities and which are corrosion current densities?
In this study, the CFRP and SPCC specimens were examined under different surface roughness conditions; Figures 2 and 3 show the corrosion characteristics of the specimens .
Figure 2 and figure 3 show the results of potentiodynamic corrosion under different surface roughness conditions for the CFRP and SPCC specimen, respectively.
The corrosion current density values for both the specimens were measured through Tafel plots that were further validated and compared with the corrosion current density values in the actual graph.
* Figures 2, 3 can be represented as a single figure 2 (a and b).
We are enough understanding of the point you pointed out. However, in this study, graphs have been grouped and organized for each test piece to confirm the corrosion characteristics by conditions for different surface roughness on the same test piece. It would be appreciated if you could understand that the corrosion behavior in the same sample was observed for each surface roughness condition of each test piece.
*The experimental data presented in table 2 must be interpreted and explained accordingly. CFRP exhibits poor electrical conductivity but its surface presents increasing and/or decreasing oxygen content before/after corrosion! WHY?
CFRP exhibits poor electrical conductivity because the surface is treated with polymer, however the corrosion rate is related to increase due to the exposure of carbon fiber surfaces by the surface roughness.
The chemical structure of epoxy molecule contains a variety of organic groups including aromatic rings such as, -CH3 and ≡C-O-C≡.
The amount of oxygen is high on the surfaces of the as-received CFRP specimen and the #2000-sandpapered specimen because of the change of aromatic rings on the epoxy surface due to the current flow through the 5.0 wt.% NaCl aqueous solution in the precipitated state. Therefore, it may be concluded that the variation of CFRP chemical composition, particularly the epoxy material, is related to the dissolution in 5.0 wt.% NaCl aqueous solution during potentiodynamic corrosion.
However, in case of the #200-sandpapered specimen, the exposed carbon fiber acts as a conductor to the current, thereby inhibiting the elution (decomposition) of epoxy, compared with the other CFRP specimens. It is judged that the change appears differently.
SPCC composition after corrosion indicates the presence of Ti, Cr, Cu, As, Zr, Nb? How is it possible that these elements appear as a result of corrosion processes? If the steel contains these elements then table 1 do not corresponds to the reality.
In the table 1 has summarized on the chemical composition for the officially SPCC by spectroscopic analysis, and its EDS analysis result, and the EDS analysis result of CFRP.
*Figures 5, 6 can be represented as a single figure 4 (a and b).
Figure 4 shows the microstructure of CFRP and SPCC before and after corrosion testing, while Figures 5 and 6 show the galvanic corrosion of CFRP and SPCC specimens with different surface roughness, respectively. We appreciate your suggestion; however, we intend to maintain these as separate figures.
*3.4. Section refers to galvanic corrosion. Figures 5 and 6 represents the chronopotentiograms of CFRP-SPCC couple and consequently indicate the Icouple variation with time (200 s). How did the authors determine the ”corrosion amount” or ”corrosion rate” from these graphs?
This study observed the galvanic corrosion tendency of CFRP and SPCC specimens with different roughness separately at the same time in the same 5 wt.% NaCl solution, rather than performing a galvanic corrosion test on a coupled test piece.
The results have been obtained by analyzing the galvanic corrosion tendency based on the surface roughness. It was confirmed that when the slope of the graph has a positive slope (indicating the increase of current during reaction time), the corrosion rate increases.
Also, the amount of corrosion is according to the value of the current measured during galvanic corrosion. As the current value increases, the volume of corrosion increases.
*Some of the references are not written in accordance with the journal's requirements.
All the references have been formatted according to journal guidelines.
*There are many grammar and typing mistakes. The authors must revise the entire manuscript.
The entire manuscript has been revised for language, grammar, and improved clarity.

Reviewer 2 Report
It is a piece of experimental record rather than a manuscript. The figures all over the manuscript are obscure, the resolutions are far from the publishable requirement. The SEM and EDS analysis are used very very often in the related area but have never seen this kind of organization, very unprofessional, please re-organized them according to the published papers.
Author Response
We have studied the potentiodynamic corrosion for CFRP and SPCC's different surface roughness conditions, and we have galvanic corrosion of CFRP and SPCC with a flat cell system in NaCl solution. And then finally, we have a review of the microstructure and chemical composition variation between potentiodynamic corrosion behaviors.
We thank all the reviewers for the constructive feedback on our manuscript. We have integrated all the suggestions and revised the manuscript accordingly. We hope that the refined manuscript will be considered for publication.
We have sorry for the reviewers. As we revised the paper, there was a need for a title change, so in “Effects of surface roughness on the electrochemical properties and galvanic corrosion behavior of CFRP and SPCC in automotive steel materials,” We would like to change to "CFRP and SPCC alloy." We ask for reviewers' understanding of this.
We would attach to the revised paper for your recommendations.
* It is a piece of experimental record rather than a manuscript. The figures all over the manuscript are obscure, the resolutions are far from the publishable requirement. The SEM and EDS analysis are used very very often in the related area but have never seen this kind of organization, very unprofessional, please re-organized them according to the published papers.
We have a summary that the resolution of the graph reviewer' mentioned has been adjusted and rearranged.
Also, the magnification of the SEM image has been reorganized into two types, X50 and X1000, include a scale bar.
Figure 4 is an observation of the microstructure of the corroded surface before and after potentiodynamic corrosion. In the case of CFRP, there is no effect on the microstructure change even after corrosion. It is related in Table 3 that the composition variation of CFRP has little effect.
However, in the case of SPCC, it can be confirmed that microstructure variates before and after corrosion occur significantly. For clarity, we tried the microstructure variates that were observed with displayed at low (X50) and high (X1000) magnifications. We have confirmed that the variation in microstructure is related to the change in the composition change of SPCC in Table 3.

Reviewer 3 Report
Dear Authors, I have a few comments for your work:
- Lines 74 to 76: "The microstructure of the as-received and surface polished specimens before and after potentiodynamic anodic polarization testing was analyzed using field emission scanning electron microscopy (FE-SEM; JSM- 7100F, JEOL, Japan) at a magnification of 50 times." Should a scanning microscope have to be used for x50 magnification?
- Chapter: 3.2. Electrochemical properties In my opinion, a table with sample parameters should be attached to all tested samples. It should be there: corrosion potential, breakdown potential, repassivation potential, transpassivation potential, polarization resistance, corrosion current and standard deviation (STD).
- Table 2 Surface composition of CFRP and SPCC before and after corrosion testing Table 2 should be edited in line with the journal's requirements.
- The analysis of corrosion products by the EDS system is not accurate. The tested surface is characterized by high porosity, therefore the measurement error is large. Keep it in mind. (Tab. 2)
- Figure 4. Microstructure of CFRP and SPCC before and after corrosion testing. Pictures can be enlarged.
In the text of the article you write 50x magnification, but there is no scale in the photos.
- Editing errors are marked in the appendix. Please correct them. The dot should be at the end of the sentence.
Yours faithfully

Author Response
We have studied the potentiodynamic corrosion for CFRP and SPCC's different surface roughness conditions, and we have galvanic corrosion of CFRP and SPCC with a flat cell system in NaCl solution. And then finally, we have a review of the microstructure and chemical composition variation between potentiodynamic corrosion behaviors.
We thank all the reviewers for the constructive feedback on our manuscript. We have integrated all the suggestions and revised the manuscript accordingly. We hope that the refined manuscript will be considered for publication.
We have sorry for the reviewers. As we revised the paper, there was a need for a title change, so in “Effects of surface roughness on the electrochemical properties and galvanic corrosion behavior of CFRP and SPCC in automotive steel materials,” We would like to change to "CFRP and SPCC alloy." We ask for reviewers' understanding of this.
We would attach to the revised paper for your recommendations.
* Lines 74 to 76: "The microstructure of the as-received and surface polished specimens before and after potentiodynamic anodic polarization testing was analyzed using field emission scanning electron microscopy (FE-SEM; JSM- 7100F, JEOL, Japan) at a magnification of 50 times."Should a scanning microscope have to be used for x50 magnification?
The SEM images were obtained at ×50 and ×1000 magnifications. We have a summary that the SEM image has been rearranged into two types, X50 and X1000 include a scale bar.
Figure 4 is an observation of the microstructure of the corroded surface before and after potentiodynamic corrosion. In the case of CFRP, there is no effect on the microstructure change even after corrosion. It is related in Table 3 that the composition variation of CFRP has little effect.
However, in the case of SPCC, it can be confirmed that microstructure variates before and after corrosion occur significantly. For clarity, we tried the microstructure variates that were observed with displayed at low (X50) and high (X1000) magnifications. We have confirmed that the variation in microstructure is related to the change in the composition change of SPCC in Table 3.
*Chapter: 3.2. Electrochemical properties In my opinion, a table with sample parameters should be attached to all tested samples. It should be there: corrosion potential, breakdown potential, repassivation potential, transpassivation potential, polarization resistance, corrosion current and standard deviation (STD).
We were thanks you pointed out, the table 2 has been reedited to suit the journal's requirements.
*Table 2 Surface composition of CFRP and SPCC before and after corrosion testing Table 2 should be edited in line with the journal's requirements.
Table 2 has been formatted according to journal guidelines.
*The analysis of corrosion products by the EDS system is not accurate. The tested surface is characterized by high porosity, therefore the measurement error is large. Keep it in mind. (Tab. 2)
Thank you for your point. The EDS results (Table 3) related that the SEM results (Figure 4). The chemical compositions of the CFRP and SPCC are given in Table 1. Nevertheless, the analysis of the chemical composition of SPCC after corrosion confirms the existence of other metallic trace elements. The reason is that there are existing metallic trace elements in the general iron ore melting process. However, because no trace elements were detected through the spectroscopic analysis of pristine SPCC, the elements have not been specified. Besides, metallic trace elements have been not detected in commercial SPCC via EDS; this can be explained by the presence of a stable oxide film of Fe2O3 on the SPCC surface. After the potentiodynamic corrosion experiment, it is determined that SPCC's metallic trace elements were detected as a result of EDS analysis for the SPCC.
*Figure 4. Microstructure of CFRP and SPCC before and after corrosion testing. Pictures can be enlarged. In the text of the article you write 50x magnification, but there is no scale in the photos.
We reorganized the SEM images in Figure 4 at 50x and 1000x. Below all the SEM images, a scale bar for each magnification has been entered. We have indicated that the 50x magnification has a 500㎛ scale bar, and the 1000x magnification has a 25㎛ scale bar..
Thank you for the reviewer's help in revising the paper.

Round 2
Reviewer 1 Report
Review on Manuscript ID: materials-916105
Title: Effects of surface roughness on the electrochemical properties and galvanic corrosion behavior of CFRP and SPCC alloy
Authors: YoungHwan Kim, MyeongHan Yoo, MinSeok Moon* The authors have taken into account the observations and consider that they have considerably improved the quality of the article. I congratulate the authors for their workAuthor Response
As requested by the reviewer, we have attached the final modified paper that Editage, an English proofreading company, edited the revised paper.
We upload the final edited file.
Could you check, please?
Thank you & Best Regards.
Min Seok Moon.

Reviewer 2 Report
- The introduction does not explain the application background of SPCC, and why choose the combination of SPCC and CRFP as the research object?
- The results in Table 1 were obtained by EDS, as a research paper, the analysis results of raw materials are not convincing.
- The properties and behaviors of materials are often affected by ambient temperature in the process of practical application, however, the author does not take this fact into account, this will overshadow the value of the study.
- Overall, the analytical methods and the presented research results used in this study are poor.
- After revising, the author has not made any radical changes in the previous version, it's still an experimental report.
Author Response
- The introduction does not explain the application background of SPCC, and why choose the combination of SPCC and CRFP as the research object?
Thank you for pointing this out. SPCC, which was used in this study, is a general-purpose steel material currently applied in all automobiles. This study identified potentiodynamic corrosion behavior at different surface roughness conditions of each SPCC and CFRP material (as these materials likely to be used in the automotive industry in the future).
In this study, the effect of surface roughness on corrosion behavior was studied by identifying the galvanic corrosion behavior under all surface roughness conditions of SPCC materials based on the best and worst surface roughness conditions of CFRP.
- The results in Table 1 were obtained by EDS, as a research paper, the analysis results of raw materials are not convincing.
Thank you for making this observation. In the initially submitted manuscript, Table 1 indicated the general alloy components of SPCC material. However, a reviewer instructed that the content of CFRP material be included. As you may well know, there are no studies with information on the component analysis of CFRP. Most studies only include component analysis information on carbon fibers. However, as this study used materials purchased from eBay, we could not confirm information regarding carbon fibers. Therefore, in adherence to the instructions of other reviewers, the EDS results obtained in this study were presented in Table 1.
Moreover, the reviewers who advised us to revise Table 1 appeared to be satisfied as they reviewed the revision paper and did not raise any concerns regarding Table 1 in their response. Therefore, we think that the other reviewer wants to more detailed information indicated for the experimental materials.
- The properties and behaviors of materials are often affected by ambient temperature in the process of practical application, however, the author does not take this fact into account, this will overshadow the value of the study.
Thank you for raising this concern. This experiment evaluated potentiodynamic and galvanic corrosion behaviors based on changes in the surface roughness of each material at 25°C. The focus of this study was that most automobile manufacturing sites work in environments that maintain 25°C, and saltwater test conditions are tested at 25°C. Therefore, the reviewer’s suggestion will be considered in further research in the future.
- Overall, the analytical methods and the presented research results used in this study are poor.
This study used the potentiodynamic corrosion analysis technique used to examine the corrosion behavior of general metal materials. There is no known corrosion analysis method for CFRP, so the same analysis technique was used. It is a case where corrosion analysis was performed in the same way in References 7 and 9.
In addition, galvanic corrosion behavior under each surface roughness condition for the SPCC and CFRP materials used in this study was established.
Generally, the galvanic corrosion was tested in a coupled state whereby the materials tested were connected by welding, bolting, or riveting. However, in this study, galvanic corrosion behavior was observed for 10 mm2 of the exposed area of each material using a double flat electrode cell. The galvanic corrosion behavior of each of the materials used in the experiment at different surface roughness conditions was identified.
Moreover, in this study, microstructural and composition changes resulting from potentiodynamic corrosion based on the surface roughness conditions of each material were determined and considered as results. A large change was observed in the components of steel materials before and after corrosion. In our opinion, only Fe and O are detected by Fe2O3, which is a stable oxide film on the surface, as the components before and after corrosion are minimal, that existed inside the molten metal during the melting process in the furnace, the process of manufacturing SPCC. Various metal components were detected as trace metal components present in the SPCC manufacturing process. Then the stable oxide film that was initially formed in the process of corrosion was eluted, and existing trace metal components were oxidized using the electrolyte.
- After revising, the author has not made any radical changes in the previous version, it's still an experimental report.
Thank you for raising this point. In general, the focus of potentiodynamic corrosion and galvanic corrosion studies is on the correlation of the corrosion behavior of a material to the theoretical background of the actual corrosion behavior of experimental material. Commonly, many studies for the potentiodynamic corrosion and galvanic corrosion have mainly been conducted using materials with excellent surface roughness conditions.
However, in this study, we have focused on the examination of the actual potentiodynamic and galvanic corrosion behaviors of the SPCC and CFRP at each different surface roughness conditions. We confirmed the results of the surface roughness is affected through the different corrosion rate.
